# Challenges, Concerns, and Experiences of Community-Dwelling Older Women with Chronic Low Back Pain—A Qualitative Study in Hong Kong, China

**DOI:** 10.3390/healthcare11070945

**Published:** 2023-03-24

**Authors:** Tiffany H. T. Wong, Kaden S. K. Lee, Sharon M. C. Lo, Mandy M. P. Kan, Crystal Kwan, Emmanuelle Opsommer, Shahnawaz Anwer, Heng Li, Arnold Y. L. Wong, Veronika Schoeb

**Affiliations:** 1Department of Rehabilitation Sciences, The Hong Kong Polytechnic University, Hong Kong SAR, China; 2Department of Applied Social Sciences, The Hong Kong Polytechnic University, Hong Kong SAR, China; 3School of Health Sciences (HESAV), University of Applied Sciences and Arts Western Switzerland (HES-SO), 1011 Lausanne, Switzerland; 4Department of Building and Real Estate, The Hong Kong Polytechnic University, Hong Kong SAR, China

**Keywords:** chronic low back pain, older people, coping strategies, thematic analysis

## Abstract

Background and Objectives: Although chronic low back pain (CLBP) is known to negatively affect multiple aspects of the lives of older people, prior qualitative studies mainly focused on the lived experiences of older people with CLBP in Western countries. Given cultural and contextual differences and poor understanding of CLBP in older women with CLBP, it is important to better understand the concerns and lived experiences of Chinese older women with CLBP. The current study aimed to investigate the experiences, challenges, concerns, and coping strategies of older women with CLBP in Hong Kong. Research Design and Methods: A total of 15 community-dwelling older women with CLBP aged ≥60 years were recruited from a physiotherapy clinic or a community center for semi-structured interviews. The interviews were audio recorded and transcribed ‘verbatim’. The transcription was imported to NVivo 12 software. Thematic analysis was conducted using Braun and Clarke’s method. Results: Five themes were identified: (1) physical impacts of CLBP on daily life; (2) psychological influences of CLBP; (3) management of CLBP; (4) family support; and (5) social activities and support. Discussion and implications: Negative physical and psychosocial impacts of CLBP were common among older women, and they adopted diverse pain management strategies, although some of their treatment options were influenced by the Chinese culture. Misbeliefs and responses of family and friends also affected their management strategies. Elderly community centers are a significant source of social support for older women with CLBP, making it an ideal platform for establishing self-help groups to facilitate their self-management of CLBP.

## 1. Introduction

Chronic low back pain (CLBP) is a debilitating musculoskeletal condition that leads to high treatment costs [1]. Approximately 85% of CLBP cases have unknown causes and are diagnosed as non-specific CLBP [2]. The reported prevalence of CLBP is 32.9% in developed countries [3], and the prevalence of CLBP increases with age [4,5]. Compared to younger adults, older adults aged 60 years older have nearly double the risk of developing CLBP partly due to more comorbidities and lower pain tolerance [6,7,8]. The estimated 12-month prevalence of CLBP in people aged 60 years or above ranged from 29.1% to 67% [9,10].

The presence of CLBP may interfere with older adults’ activities of daily living (e.g., housework, walking, standing, transfer, or even sitting) [11]. If there are neurological involvements, these patients may experience radiculopathy, lower limb muscle weakness, and aberrant sensation or reflex, which further reduce their functional mobility [12]. Additionally, CLBP may cause psychological and emotional distress in older people (e.g., fear avoidance beliefs, depression, and anxiety) [11]. Research studies revealed that older adults with higher persistent pain intensity were more likely to experience psychological distress [13,14]. CLBP-related restrictions in daily activities also hinder older adults in fulfilling their social or caring role within their families [12]. Notably, their roles may shift from care-providers to care-receivers, which may affect their self-esteem. Physical and psychosocial impacts of CLBP are interrelated, as such, CLBP may lead to reduced quality of life and increased reliance on healthcare services among older adults [15].

Although multiple quantitative studies and reviews have attempted to identify factors associated with CLBP in older adults [16,17,18,19], these studies only collected data from self-reported questionnaires. The in-depth concerns or lived experiences of older adults with CLBP are difficult to uncover through questionnaires [20]. A more comprehensive understanding of concerns and needs of older people can facilitate health resources allocation and social services planning. A thorough understanding of feelings and lived experiences of older adults with CLBP in a local community could help healthcare providers and policymakers develop more relevant health and social policy to support older people with CLBP [20]. Qualitative studies are well-suited for achieving such aims.

Several qualitative studies from different countries have investigated the lived experiences in older adults with CLBP [15,21,22,23,24,25]. These studies revealed that back pain reduced the independence of older adults, causing increased frustration [15]. Additionally, older adults might prefer new treatments to conventional treatments [23], and their maladaptive beliefs might lead to maladaptive coping strategies [25]. However, Chinese older adults with CLBP may have different lived experiences given the differences in culture and healthcare systems. Importantly, no qualitative research has investigated the concerns and needs of community-dwelling older Chinese women with CLBP who are more likely to experience CLBP [26], but are easily overlooked by researchers. Therefore, this study aimed to: (1) examine the challenges, concerns, and experiences associated with CLBP and the respective coping strategies among older women who are living alone or with families in Hong Kong; and (2) understand the pain management among Chinese older women with CLBP in Hong Kong.

## 2. Materials and Methods

This study was approved by the Human Subject Ethics Committee of the Hong Kong Polytechnic University (Reference no: HSEARS20210128001-01) and was conducted according to the Declaration of Helsinki. Semi-structured interviews were conducted to understand the lived experience and needs of older women with CLBP. The current study followed the consolidated criteria for reporting qualitative research (COREQ) [27].

### 2.1. Participants

Participants were recruited by physiotherapists in physiotherapy clinics or social workers in an elderly community center between April 2021 and February 2022 by convenience sampling. Community-dwelling older women aged 60 years or older who lived in community settings outside nursing homes were eligible for this study if they met the following criteria: (1) had CLBP along or near the lumbosacral region with or without leg pain that persisted for at least 3 months in the last 12 months 19; (2) could communicate in Mandarin, Cantonese, or English; and (3) were self-ambulatory in the community with or without walking aids. Individuals with cognitive impairment, severe psychiatric illness, neurological disorder, undergoing prior lumbar surgery, and/or cancer were excluded.

### 2.2. Interviews

All semi-structured interviews were conducted either on a university campus or in an elderly community centre. One online interview was arranged for a participant who was worried about meeting people in person during the COVID-19 outbreak. An experienced qualitative researcher (AW) who is a male physiotherapist led two trained research assistants (CW and HC) and two physiotherapy students (SL and TW) who conducted the interviews using an interview guide (Appendix A). Each interview lasted for approximately one hour. After explaining the objective of the experiment and obtaining informed consent from the participant, the interviewer gave a standardized introduction and collected socio-demographic data, followed by asking open-ended questions related to the following: (1) CLBP experiences; (2) impacts of CLBP on various aspects of life (e.g., family and friends); (3) source of knowledge of CLBP; and (4) coping strategies for CLBP. All interviews were audio-recorded, and the recordings were stored in a password-protected laptop, which was only accessible to authorized research personnel. A participant code was assigned to each participant to ensure anonymity.

### 2.3. Data Analysis

Four bilingual researchers (CW, TW, SL, and KL) transcribed ‘verbatim’ in Chinese, and then translated into English. Thematic analysis was conducted according to the six steps suggested by Braun and Clarke [28]. All transcripts were imported to NVivo 12 software (released March 2020). Five bilingual researchers became familiar with the data by reading all transcripts word-for-word. One researcher (CW) classified the texts of three transcripts into meaningful codes to create an initial codebook. Another three researchers (KL, SL, and TW) then verified and updated the codes and codebook. The fifth researcher (MK) independently verified the codes in the text and codebook. Four researchers (KL, SL, TW, and MK) then categorized the codes into themes and subthemes. The coding procedure was repeated for the subsequent transcripts. Data saturation was reached when no more new themes were identified from an additional interview. To enhance the rigor and trustworthiness of the findings, all team members discussed the categories to develop themes until a consensus was reached.

## 3. Results

A total of 15 community-dwelling older women were recruited (Table 1) after approaching 16 eligible participants. Of the participants, 13 were aged 71 years or older. Eight of them were married, seven were widows, and one was single. Twelve were living with family, relatives, or domestic helpers, while four were living alone (Table 1). Five themes, seven subthemes, and two nested subthemes were identified (Figure 1). Relevant quotes of each theme are presented in Appendix B.

### 3.1. Theme 1—Physical Impact on Daily Life

CLBP affected the participants’ activities of daily living. The participants reported difficulties in walking/sitting/standing (*n* = 11), taking stairs (*n* = 2), doing exercises (*n* = 3) or housework (*n* = 12), or carrying heavy objects (*n* = 8). Some had difficulties in sleeping (*n* = 10).


*“I used to be fine because I did regular exercises, such as swimming… After I have LBP, I hate swimming and try to avoid it… my back was like hit by water from a waterfall. It was too painful for me to continue swimming.”*
(H5)


*“When I sleep, my back will be in pain. …Sometimes I suffer from insomnia. …I can’t sleep at night, so I sleep in the morning. My days and nights are reversed…”*
(A4)


*“I couldn’t sleep well, and my back was in pain.”*
(A6)


*“I couldn’t sleep for two whole nights in a month. I feel painful when I have slept for a while, so I don’t want to sleep.”*
(H2)

### 3.2. Theme 2—Psychological Impacts of CLBP

#### 3.2.1. Subtheme—Emotion

Fourteen participants experienced different extents of LBP-related psychological disturbance, ranging from low mood to depressive symptoms. Loneliness, sadness, irritation, annoyance, frustration, and unhappiness were also reported.


*“I feel…sometimes desperate because of the incurable condition”*
(A6)


*“It’s exhausting and lonely to suffer from the back pain.”*
(H3)

Most of them indicated their worries about losing their functional independence (*n* = 6) and not being able to support their families (*n* = 2).


*“I am worried that I may not be able to walk anymore.”*
(C2)

Only one participant expressed that her mood was not affected by LBP nor worried about it. Another participant mentioned how her outlook changed after accepting her pain.


*“I am not depressed now; it has been there for many years, and I have accepted it.”*
(A5)

#### 3.2.2. Subtheme—Attitudes toward CLBP

Some older people thought that their pain would stay with them for the rest of their lives. Therefore, they felt hopeless and helpless. They tended not to disclose their situation to others because they thought that it might increase others’ burden, and others could not help them.


*“… I feel like this…injury will stay with me for the rest of my life.”*
(A3)


*“I don’t want to increase their [referring to her family] burden. Besides, there’s nothing they can do to help me.”*
(C1)

One participant said that even if she shared her situation with her family members, she still could not get any help from them,


*“... They thought that it was something simple. … Nobody helped me, I did everything by myself.”*
(C3)

### 3.3. Theme 3—Management of CLBP

#### 3.3.1. Subtheme—Self-Management of CLBP

To cope with CLBP, participants used diverse management strategies, such as exercise (*n* = 12), pain relief plasters (*n* = 8), medicated oil (*n* = 8), Chinese medicine (*n* = 7), painkillers (*n* = 6), ointment (*n* = 5), and massage (*n* = 3).

Some participants believed that exercise was important and effective in reducing pain and preventing further deterioration of the back. However, some deemed that exercise was not helpful in pain relief.


*“I may forget the pain after exercising. …I tend to do more exercise. … I keep on doing it as I hope my back won’t deteriorate quickly.”*
(A1)


*“I have been doing exercise, but the pain has not been relieved.”*
(C3)

#### 3.3.2. Subtheme—Seeking Help from Clinicians

##### Nested Subtheme—Treatment Choices and Effectiveness

The participants received LBP treatments from different clinicians, including general practitioners (GPs) (*n* = 11), Chinese medicine practitioners (*n* = 10), physiotherapists (*n* = 9) and acupuncturists (*n* = 8), orthopedists (*n* = 5), bone setters (*n* = 4), chiropractors (*n* = 3) and Tui Na therapists (*n* = 2). (Tui Na is an ancient form of massage that is a fundamental part of traditional Chinese medicine. “Tui” means pushing and “Na” means grasping.) Four participants received more than four types of treatments. Another eight participants received three to four types of treatments. Although most of the participants visited GPs for their CLBP problems, some participants shared that there was little impact.


*“I have visited different doctors…but my back hasn’t got any better. …”*
(A2)


*“Although I have visited a doctor, I didn’t receive any specific treatment. I only received some heat therapies, but they were useless.”*
(H2)

Some participants preferred Chinese medicine practitioners to GPs because the former holistically managed their condition. However, the beneficial effects of acupuncture varied among the participants.


*“… I hate taking medicine, but I tend to accept Chinese medicine instead of western medicine. I think Chinese medicine helps me regulate my body. … sometimes after I took the western medicine, my mouth would be dry and uncomfortable. …”*
(A5)


*“I spent around HK$200 to HK$300 [approximately US$25 to US$38] to receive acupuncture each time. However, it’s not effective.”*
(H3)


*“I received acupuncture from my son-in-law, I feel much relieved.”*
(A6)

Some participants found that physiotherapy was effective in decreasing pain and improving function, but they could not receive physiotherapy treatment continuously because of limited resources in public hospitals.


*“After receiving physiotherapy, I realized that there were some movements that I could do. These could be very useful. I think physiotherapy was quite effective… They [GPs] said, the number of times to receive physiotherapy was limited.”*
(A3)

###### Nested Subtheme—Communication with Healthcare Professionals

Ten participants had a good rapport with healthcare professionals. They reported that they encountered medical personnel who were caring and helpful.


*“… they [the medical doctors who met A3] were helpful. They taught me some movements and … explained how these movements would be useful to me.”*
(A3)

However, most of the participants commented that the GPs only prescribed painkillers without sufficient explanation or showed a lack of understanding of patients and provided insufficient information (*n* = 6). They commented that Chinese medicine practitioners were more caring than GPs.


*“My GP (…) gave me some painkillers and ointments. He didn’t give me enough time to talk. There was no examination either. (…) My Chinese medicine practitioner took my pulse. He would check if there were problems associated with ‘blood deficiency’ or old age. Then he prescribed Chinese medicine to regulate my body. (…) I don’t think the GP cared about me (…) I think the Chinese medicine practitioner is better.”*
(C1)

### 3.4. Theme 4—Family Relationships

#### 3.4.1. Subtheme—Support from Family Members

All participants received different kinds of family support, including financial/material support (*n* = 11), informational support (*n* = 9), social companionship (*n* = 4), and psychological support (*n* = 3).


*“My children were worried and brought me to visit two to three doctors (…) My children treat me very well and they give me sufficient care (…) My children paid for the medical expenses (…) My children hired a part-time domestic helper. (…) My children arranged my treatment.”*
(H1)


*“We [participant and her children] go to Chinese restaurants together when they have holidays.”*
(C4)

One participant whose family members were physiotherapists received regular treatments from them.


*“They even gave me acupuncture. (…) They like to give me physiotherapy.”*
(A6)

#### 3.4.2. Subtheme—Poor Family Bonding

However, some participants experienced insufficient family support because their family members were too busy, or they only gave general advice or unhelpful suggestions to the participants.


*“My children seldom visit me because they are busy. (…) I seldom discuss my things with others. I don’t want to increase their burden. Besides, there’s nothing they can do to help me. (…) It’s not necessary to talk about this.”*
(C1)


*“They just reminded me to take a rest on my bed when I am in pain. They believed I should move less and only perform the movements when I feel good.”*
(A3)

One participant even commented that her family members, who lived with her, would not notice whether she had died in bed.


*“If I die in my bed, he [participant’s son] won’t even realize (…) One time I fell and felt dizzy. I went to the washroom to vomit. My son asked ‘Mum, what’s the matter?’ I said, ‘I’m exhausted, I hit my head and it’s painful, I’m also vomiting.’ My son replied, ‘Go to see a doctor!’, and then he left the washroom”*
(A4)

### 3.5. Theme 5—Social Activities and Support

#### 3.5.1. Subtheme—Social Activities and Support from Friends

Participants received different kinds of support from their friends through gatherings in Chinese restaurants or churches, phone calls, advice on pain management, and the provision of support for their daily life.


*“Sometimes I will share my CLBP condition with the friends I met in the Chinese restaurant.”*
(A6)


*“They [Participant A3’s friends] told me that some movements could be useful to me and encouraged me to try.”*
(A3)


*“Sometimes I call my friends and chat with them when I feel lonely. (…) They care about me.”*
(H3)


*“My friend also take care of me from time to time. My friend cooks meals for me.”*
(H5)

However, the participants reported that their social life was affected by LBP (*n* = 10). For example, they went out less frequently, preferred staying at home, and no longer enjoyed travelling as much as before. Some also went to Chinese restaurants less frequently.


*“I used to visit my relatives. Now I go out less and have fewer chances to go to Chinese restaurant with my friends.”*
(C1)

#### 3.5.2. Subtheme—Elderly Community Centre (ECC)

Many participants enjoyed participating in activities organized by an ECC. They liked to use the facilities in the ECC, such as the exercise equipment, computers, karaoke, newspapers and board games. They could also join different activities organized by the ECC (e.g., exercise classes, educational talks, and interest classes). By joining these events, the participants made more friends to enlarge their social circle and alleviate their stress.


*“From time to time, there are exercise classes here. I may forget the pain after the exercise. Luckily, there are exercise classes here for me to improve my health. (…) I rely on such classes to do exercise. (…) This friend comes from the centre. (…) I feel better when I come here, it eases my mood. This place is good, it’s helpful for me.”*
(A1)

## 4. Discussion

Most of the participants reported CLBP-related negative impacts and disturbances of daily activities. To reduce these negative impacts, they adopted different self-management approaches. However, not all these approaches were effective. The participants’ and/or their caregivers’ misbeliefs related to CLBP and poor relationships with healthcare providers might hinder them from using effective self-management strategies for CLBP. While some participants enjoyed support from families and friends, others felt insufficiently supported by their families. ECCs can be an important platform for older people with CLBP to receive proper health education and social support through various events [29,30].

CLBP not only causes functional limitations but also disturbs sleep [31]. Most of our participants reported sleep disturbance, which interfered with their daily routine and emotions. Research has suggested that poor sleep aggravated pain by sensitizing pain perception, [32] while pain can also interfere with one’s sleep quality [33], resulting in a vicious circle [34]. Additionally, sleep disturbance can be a major source of psychological distress in people with chronic pain [35]. Hester and Tang (2008) revealed that people with concurrent chronic pain and sleep disturbance had poorer physical and psychosocial functioning than those with chronic pain alone [35]. Therefore, improving sleeping quantity/quality among older adults with CLBP should be emphasized in pain management [36].

In addition to sleep disturbance, our participants expressed concerns about deteriorated functional ability and difficulty in performing exercises. They tried different approaches to reduce the negative impacts of CLBP (e.g., avoiding pain-provoking activity). However, this fear avoidance behavior might lead to the adoption of a sedentary lifestyle, which further deteriorates their physical fitness and functions [37]. Some participants preferred staying at home when their pain was intolerable. Prior research has found that chronic pain might prevent people from enjoying their pre-morbidity hobbies, [38] and might lead to social isolation, depression, and reduced quality of life [24]. Further, worries of being physically dependent may interfere with patients’ mindsets and behaviors, resulting in repetitive negative self-perception and poor psychological health [39].

Some participants preferred not to disclose their pain to others because they perceived that no one would understand their pain. A similar finding was reported by Rodrigues de Souza and colleagues [40]. Pain is an individual sensation and is difficult to be understood by others, especially for older adults who try to hide their pain [41]. Therefore, their pain may be underestimated by others [42]. The development of age-related pain assessments for older people with LBP is warranted in future research [43].

Since CLBP might not necessarily have specific causes or pathology, older adults should be reassured of the benign nature of CLBP and learn to self-manage their pain 40. Barriers to pain management (e.g., uncertain diagnosis, social stigma, ineffective treatments, inadequate clinical knowledge, and poor health literacy) should be addressed to help older adults self-manage their pain [44].

Hong Kong offers both public and private healthcare services. The public healthcare is similar to the National Health Service in the UK. The Health Bureau and the Hospital Authority are the two major stakeholders for formulating the health policy and providing the subsidized universal care in Hong Kong [45] (Schoeb, 2016). The private healthcare sector in Hong Kong provides services to those who can pay by themselves or are covered by private health insurance. Given the Chinese culture in Hong Kong, many people with CLBP see private traditional Chinese medicine practitioners or bonesetters for treatments. These practitioners provide herbal treatments and manual therapy to patients.

The most common pain management strategy adopted by the participants was exercising. Exercise is known to be the most effective LBP management strategy regardless of exercise types [46,47]. Importantly, the participants generally believed that exercise could help delay their health deterioration and maintain functional mobility. Because adequate physical activity can prevent various diseases, [48] improve quality of life, and delay functional dependency in older adults [48,49,50], clinicians should emphasize the importance of exercise to older adults with CLBP.

Good professional–patient relationships can facilitate effective healthcare service delivery and optimize treatment outcomes and patients’ satisfaction [51,52]. Our participants were highly satisfied with traditional Chinese medicine consultation. They commented that Chinese medicine practitioners were more caring than GPs. Given the concept of blood, Qi, and meridian in traditional Chinese medicine, Chinese medicine practitioners need to pay attention to all signs and symptoms of patients to identify potential disturbance in the body [53]. (To maintain individual’s health, traditional Chinese medicine emphasizes the interaction or associations among body, mind, emotions, and environment. Blood, Qi, and meridian are the key components for maintaining a harmonious environment within the body. Any disturbance in one of the components will contribute to disease or pain.) Therefore, they usually thoroughly examine patients through observations, listening, questioning, and pulse palpation, which involve many communications and interactions with patients, resulting in a favorable clinician–patient relationship [54]. Importantly, these doctors prescribe personalized medicine. Such a positive clinician–patient partnership facilitates patients sharing their experiences and learning self-management skills from clinicians [55].

Although the participants indicated that family members gave them informational support (e.g., advice on CLBP management), such advice might not necessarily be based on scientific evidence. Misinformation may lead to persistent pain and more CLBP-related disability [56,57]. For example, the misconception that pain-provoking activities can cause injury or recurrent back pain may result in the adoption of fear avoidance behavior and deconditioning. Likewise, the common belief that pain is a natural part of aging may discourage older people from self-managing CLBP [58,59].

Most of the participants did not receive sufficient esteem support from their family members. Esteem support is attained when a person is encouraged and/or accepted by others [60]. Some participants were reticent because they did not want to bother family members or believed that no one could help them. Therefore, participants living alone and with family members did not seem to have any differences in terms of their perceived esteem support. This is a common phenomenon among older people [40]. It is particularly obvious among Asian cultures that stress self-reliance; that is, individuals should be responsible for their own personal problems without bothering others; otherwise, it may damage their relationships with others [61].

On the contrary, most of the participants (including those who lived alone) received various instrumental support (e.g., financial aid, material resources, and needed services [61]) from their immediate or distant relatives. Likewise, some of them also provided support for their family. Mutual support and reciprocal obligation are part of the Confucian principles in Chinese culture [62]. Thomas’ [63] study revealed that older adults felt more independent and useful by giving support to others despite their physical challenges. As such, our participants might have a positive impact on their well-being by providing support to family members.

Almost all our participants went to an ECC and had positive experiences in social companionship, esteem support, informational support, and pain relief. Baranwal and Mishra [30] revealed that many older people went to day-care centers because they did not have sufficient social support. Day-care centers or ECCs provide comprehensive benefits to older adults and their caregivers alike [29,30]. These centers not only provide older adults with CLBP with proper social and emotional support, but also help prevent or delay the institutionalization of older adults [30]. ECCs regularly organize various health talks. As such, older adults can gain informational support from the centers. The exercise equipment, various entertainment facilities, and social workers in ECCs also provide both instrumental and psychological support to seniors with CLBP. Imperatively, ECCs allow caregivers to break from their prolonged caring duties and relieve their emotional distress [58].

### 4.1. Implications

Listening to the lived experiences of older women with CLBP provides a more fulsome understanding of their needs. An effective patient–clinician relationship could support patients to deal with physical and psychosocial limitations. Furthermore, our findings highlight the importance of forming self-help groups for older women with CLBP in Hong Kong. Currently, no relevant self-help groups are available in local ECCs. Research has demonstrated the benefits of self-help groups in enhancing health-promoting behaviors and health outcomes in people with CLBP [64,65,66]. Specifically, these groups may help improve emotional well-being and fatigue, promote self-management of pain, and mitigate functional disability and the duration/extent of disability [67,68].

Effective self-management of pain can lessen the burden on healthcare providers [68]. In addition to healthcare professionals, family members and caregivers are recommended to participate in pain self-management training [66,69]. Such training includes coping strategies to manage frustration, fatigue, pain, and isolation. Social workers can also be involved to facilitate the communications among patients, family members, friends, and clinicians to optimize clinical outcomes [68].

### 4.2. Limitations

This study has some limitations. First, nearly all our participants were recruited from an ECC. Our findings might predominantly reflect the lived experiences of relatively active older adults with CLBP. That said, people recruited by clinicians shared similar experiences as those recruited through the ECC. Future studies should recruit more socially inactive and/or isolated older adults to better understand their concerns and needs. Second, our findings cannot be generalized to older men. As older males are more socially isolated than older females [70], their concerns and needs may be very different. Future research should focus on male older adults with CLBP to understand the impacts of CLBP on males so that gender-specific support can be provided.

## 5. Conclusions

This is the first qualitative research to understand the negative physical and psychosocial impacts on the lived experiences of older Chinese women. To minimize these negative impacts and maximize functional independence, these women adopted different management strategies to cope with their pain. However, their management strategies might not be as effective due to misconception/misinformation about CLBP. Further, older women with CLBP may not obtain sufficient support from families and friends, which may further affect their self-management of CLBP. Given that good clinician–patient relationships can help educate self-management skills for older adults, clinicians should build a good rapport with older women with CLBP. Importantly, ECCs can be a good platform to establish self-help groups and provide various resources to support self-management of CLBP among older women so that they can age in the community.

## Figures and Tables

**Figure 1 healthcare-11-00945-f001:**
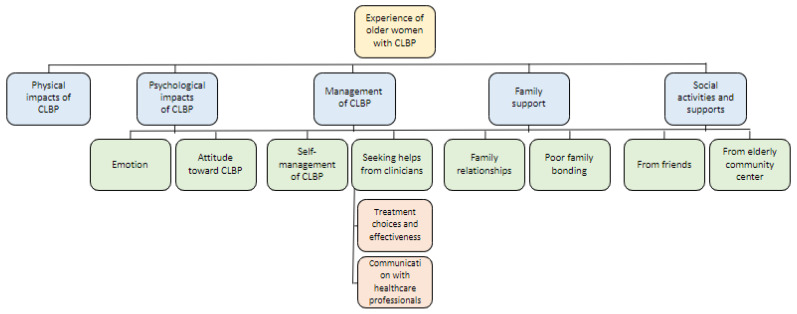
Analytical themes, subthemes, and nested subthemes. CLBP = chronic low back pain.

**Table 1 healthcare-11-00945-t001:** Demographic information of participants.

	Number of Participants	Percentage
**Age (years)**		
60–70	2	13.3
71–80	7	46.7
80–90	6	40
**Education**		
Primary school or below	8	53.3
Secondary school	6	40
University	1	6.7
**Marital status**		
Single	1	6.7
Married	8	53.5
Widow	6	40
**Household members (living condition)**		
With spouse	5	33.3
With children	2	13.3
With spouse and children	2	13.3
With relatives	1	6.7
With domestic helper	1	6.7
Alone	4	26.7
**Types of housing**		
Public housing apartment	13	86.7
Self-own apartment	2	13.3
**History of pain (years)**		
1–10	6	40
11–19	5	33.3
>20	4	26.7

## Data Availability

Relevant data supporting reported results can be made available from corresponding author on request.

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
