# Peer review of "Challenges, Concerns, and Experiences of Community-Dwelling Older Women with Chronic Low Back Pain—A Qualitative Study in Hong Kong, China"

_healthcare, 2023, doi:10.3390/healthcare11070945_

Round 1

Reviewer 1 Report

The manuscript “Challenges, Concerns, and Experiences Of Community-Dwelling Chinese Older People with Chronic Low Back Pain – A Qualitative Study” is significant for understanding the experience of older adults on musculoskeletal disorders. I have following suggestions. The current study explored the experiences, challenges, concerns, and coping strategies of older adults with CLBP in Hong Kong.

Title

- Should add Hong Kong in the title.

- Suggested title: “Challenges, Concerns, and Experiences of Older Women with Chronic Low Back Pain – A Qualitative Study in Hong Kong, China”

- Remove gender from the table, as all of your participants are female. (Table 1. Demographic information of participants).

Introduction

- Why is your study among women? Provide justification.

A similar study recently conducted “Schoeb, V., Misteli, M., Kwan, C., Wong, C. W., Kan, M. M., Opsommer, E., & Wong, A. Y. (2022). Experiences of community-dwelling older adults with chronic low back pain in Hong Kong and Switzerland–A qualitative study. Qualitative pain research: Capturing and integrating cultural, social and linguistic data, 16648714, 82.” DOI: 10.3389/fresc.2022.920387

- How did your study differ from the above study?

Methods

Should mention about COREQ, For example:

Tong, A., Sainsbury, P., & Craig, J. (2007). Consolidated criteria for reporting qualitative research (COREQ): a 32-item checklist for interviews and focus groups. International journal for quality in health care, 19(6), 349-357. https://doi.org/10.1093/intqhc/mzm042

Results

Add a table on the coding tree – theme, sub-themes and codes (modify Table 2. Analytical themes and subthemes)

Discussion

The discussion section needs to be improved

- What about their overall care pathways?

- What about traditional/alternative medicine used to manage Chronic Low Back Pain in your settings?

- How did the older adults living alone manage their health? For example:

Mahapatra, P., Sahoo, K. C., Desaraju, S., & Pati, S. (2021). Coping with COVID-19 pandemic: reflections of older couples living alone in urban Odisha, India. Primary Health Care Research & Development, 22, e64. DOI: 10.1017/S1463423621000207

Reviewer 2 Report

Minor revisions:

1)      Your abstract mentions that your subject included n=15 older women over the age of 60. However, your materials and methods section mentions only ‘older adults’. Your results confirm that it is indeed n=15 older females. Could you please clearly mention that in your materials and methods as well.

2)      Also – as your article is based on older women only – your title is misleading.  Please change it to ‘Challenges, Concerns, and Experiences Of Community-Dwelling Chinese Older Women with Chronic Low Back Pain – A Qualitative Study

3)      It appears from your article that the strength of the bond shared within families (theme 5) is also an important parameter. Lines 243-256 deserves a subtheme of ‘family bonding’ to better categorize your observations.

4)      In all your sections - You need to replace ‘older adults’  with ‘older women’ as your data is a true representation of older women.

5)      You need to re-frame your paper focused on older women to use it a s a strength rather than indicating it as a limitation. Re-write your title as suggested in point 2 and re-phrase all your results and observations focused on older women. Considering research often overlooks the impact on women – your article if framed for ‘older women’ as your target population will provide valuable data to the field.

Reviewer 3 Report

It would be better for readers to understand if the introduction is shortened a little.

Please write the year after "Osama" on line 55 in your introduction section.

Were people who underwent low back surgery included in your study?

In table 1, please mention the main activity time. I think the activity time will be helpful for your research.

You talked about community dwelling, but we need a detailed description of the living hours and living environment.
